# Long term anti-SARS-CoV-2 antibody kinetics and correlate of protection against Omicron BA.1/BA.2 infection

Javier Perez-Saez[1,2] ✉, María-Eugenia Zaballa [1], Julien Lamour [1], Sabine Yerly [3], Richard Dubos[1], Delphine S. Courvoisier[4,5], Jennifer Villers [1], Jean-François Balavoine[6], Didier Pittet[6,7], Omar Kherad[6,8], Nicolas Vuilleumier[3,6], Laurent Kaiser[3,6,9,10], Idris Guessous [11,12,23], Silvia Stringhini [1,13,23], Andrew S. Azman [1,2,23] & the Specchio-COVID19 study group*

Binding antibody levels against SARS-CoV-2 have shown to be correlates of protection against infection with pre-Omicron lineages. This has been challenged by the emergence of immune-evasive variants, notably the Omicron sublineages, in an evolving immune landscape with high levels of cumulative incidence and vaccination coverage. This in turn limits the use of widely available commercial high-throughput methods to quantify binding antibodies as a tool to monitor protection at the population-level. Here we show that anti-Spike RBD antibody levels, as quantified by the immunoassay used in this study, are an indirect correlate of protection against Omicron BA.1/BA.2 for individuals previously infected by SARS-CoV-2. Leveraging repeated serological measurements between April 2020 and December 2021 on 1083 participants of a population-based cohort in Geneva, Switzerland, and using antibody kinetic modeling, we found up to a three-fold reduction in the hazard of having a documented positive SARS-CoV-2 infection during the Omicron BA.1/BA.2 wave for anti-S antibody levels above 800 IU/mL (HR 0.30, 95% CI 0.22-0.41). However, we did not detect a reduction in hazard among uninfected participants. These results provide reassuring insights into the continued interpretation of SARS-CoV-2 binding antibody measurements as an independent marker of protection at both the individual and population levels.

While by mid-2022, a large fraction of the global population had developed anti-SARS-CoV-2 binding antibodies through infection and/or vaccination[1,2], it remains unclear whether seroprevalence results translate into the prevalence of effective protection against infection[3]. Neutralizing antibodies may provide a reliable correlate of protection against both infection and severe disease[4–7]. Neutralization assays are, however, labor-intensive and challenging to use at a large scale, despite advances in high-throughput surrogate assays[8].

Binding antibody measurements have been found to correlate with neutralization capacity against the ancestral SARS-CoV-2 strain at different degrees depending on time post infection/vaccination and on immunoassay[4,9,10]. Evidence for their more general use as a correlate of protection is mounting both from population-level[4,9], as well as individual-level studies in the context of vaccine trials[5,11–13]. These studies suggest that higher antibody titers after infection and/or vaccination tend to reduce subsequent infection risk[14]. However, most of these studies focused on the ancestral SARS-CoV-2 strain, and only a

---

A full list of affiliations appears at the end of the paper.  *A list of authors and their affiliations appears at the end of the paper. ✉e-mail: javier.perez@hcuge.ch

few reports have explored the extension of these results to the Omicron subvariants[15–18].

The evaluation of binding antibody levels as correlates of protection are challenged by the constant evolution of the anti-SARS-CoV-2 immune landscape through vaccination and successive epidemic waves driven by different virus variants. Longitudinal antibody studies up to 14 months follow-up have shown that antibody levels change with time since infection and/or vaccination across individuals and depending on the immunoassays used for detection[19–22]. Characterization of long-term antibody kinetics provides an opportunity for leveraging serological cohort studies to complement vaccine trials in evaluating binding antibody levels as a correlate of protection against future infections. By relying on binding antibody immunoassays that are simple, standardized, and widely used worldwide, the results of these studies have the potential to be generalized to other settings despite their functional limitation. These cohort studies might therefore contribute by assessing the extension of results to different commercially available immunoassays and a wider range of infection/vaccination histories, in particular to non-vaccinated individuals.

In this study, we aim to evaluate the use of a widely available immunoassay as a correlate of protection in the Omicron era. We do so by leveraging repeated serological measurements and reported infections on a population-based longitudinal cohort followed for up to 20 months in the state of Geneva, Switzerland. We first characterize antibody dynamics during the longitudinal serology period (April 2020 to December 2021) using kinetic models fit to observed antibody measurements. We then project each individual's antibody trajectories into the Omicron exposure period (December 2021 to March 2022) to explore the relationship between projected antibody levels and having a SARS-CoV-2 positive test.

## Results

The cohort included in this study was composed of 1083 adult participants recruited during the longitudinal serology period (Fig. 1), 55% of whom were female, and 91% were younger than 65 years (Table 1). Participants in the cohort had few comorbidities and no immunosuppressive diseases in general. Among participants, 91% had a history of SARS-CoV-2 infection (based on positive tests or anti-N serology as

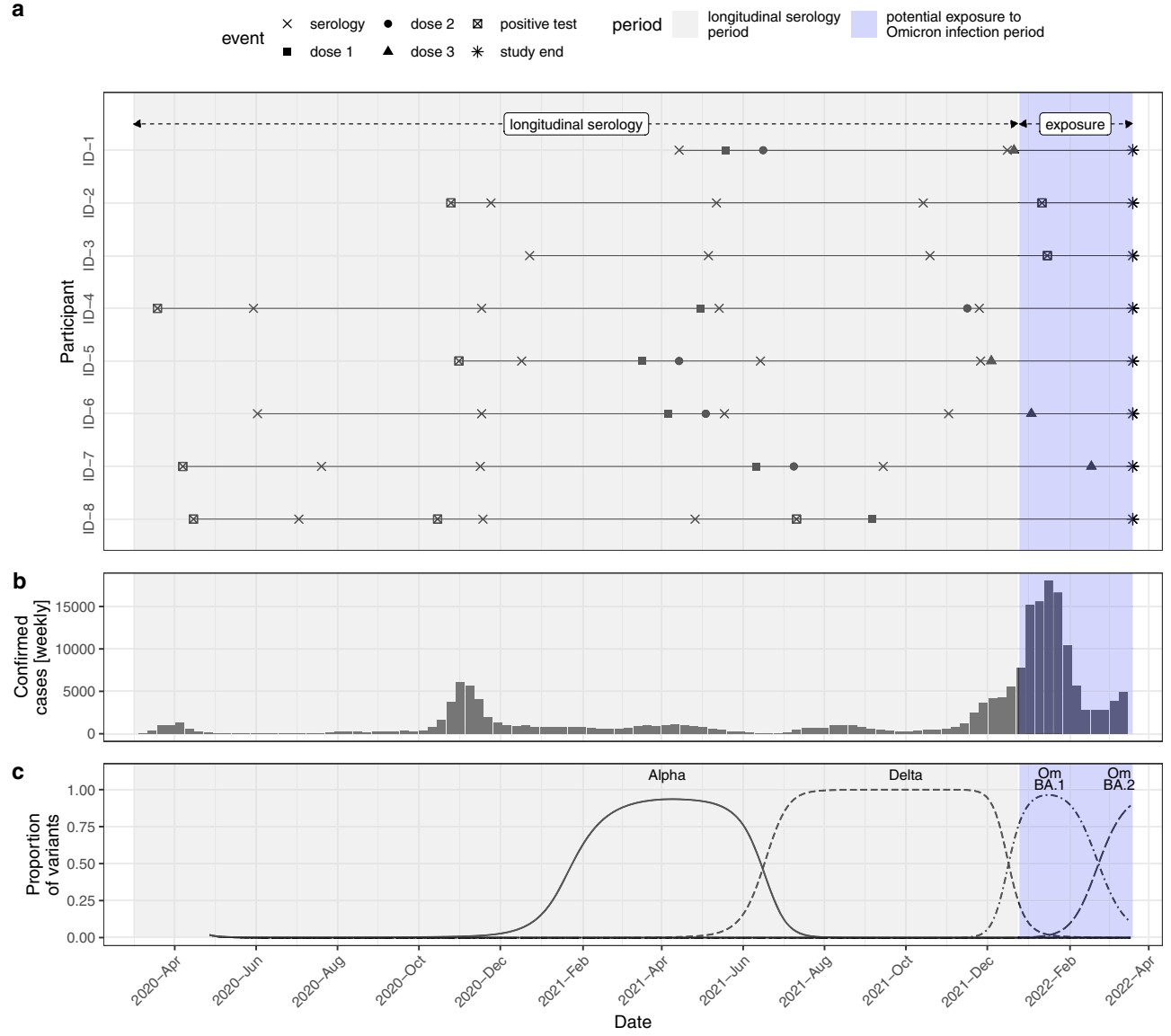

**Fig. 1 | Study context. a** Study phases and eight examples of participant-level data. **b** Weekly confirmed SARS-CoV-2 cases in the state of Geneva (available from: https://infocovid.smc.unige.ch). **c** Proportion of SARS-CoV-2 variants in sequenced samples in Western Switzerland estimated through multinomial spline regression of publicly available weekly sequence data from the Covariants project (https://github.com/hodcroftlab/covariants/), following analysis from https://www.hug.ch/laboratoire-virologie/surveillance-variants-sars-cov-2-geneve-national (report May 2022). Source data are provided as a Source Data file.

**Table 1 | Participant characteristics**

| Characteristic | N = 1083[1] |
|---|---|
| *Sex* | |
| Female | 590 (55%) |
| Male | 493 (45%) |
| *Age (years)* | |
| 18–64 | 988 (91%) |
| 65+ | 95 (9%) |
| *Infection/vaccination status at the time of most recent serological test* | |
| Infected only | 634 (58%) |
| Vaccinated only | 98 (9%) |
| Infected prior to vaccination | 310 (29%) |
| Infected after vaccination | 11 (1%) |
| Infected prior to vaccination and re-infected after vaccination | 30 (3%) |
| *Virological confirmation among participants with a history of infection (N = 985)* | |
| Virologically confirmed infection (self-reported or from ARGOS registry) | 567 (58%) |
| No virological confirmation (anti-N positive serology only) | 418 (42%) |
| *Vaccine type among vaccinated participants at the time of last serological status (N = 449)* | |
| mRNA-1273 (Moderna/US NIAID) | 247 (55%) |
| mRNA-BNT162b2/Comirnaty (Pfizer/BioNTech) | 202 (45%) |
| *# Positive Roche-S serological samples per participant within this study* | |
| 2 | 603 (55%) |
| 3 | 374 (35%) |
| 4 | 104 (10%) |
| 5 | 1 (<0.1%) |

[1]n (%)

Source data are provided as a Source Data file.

defined in the Supplementary Material Section S1), and 58% were unvaccinated at the time of their most recent serology (range of last serology dates November 16th, 2020–December 17th, 2021, median June 21st, 2021). Around half of the participants with a history of infection based on a positive anti-N serology did not have any diagnostic screening tests during the acute phase of their infection (polymerase chain reaction (PCR) or rapid diagnostic test (RDT)) (58%). Most participants in the cohort had only two positive serological tests (56%), and 10% had 4 or more seropositive samples. All event data is presented in Supplementary Fig. S2.

**Antibody trajectories during the longitudinal serology period**
Serological samples were collected from 1083 participants between April 2020 and December 2021, with follow-up times between the first and last visits of up to 20 months following infection and up to 8 months following vaccination (Fig. 2a). Over this longitudinal serology period, we did not observe anti-S-based seroreversion for any participants (Fig. 2b). Antibody levels following vaccination were distributed in the upper range of the immunoassay's dynamic range (Fig. 2b, Supplementary Fig. S3), with one in three samples collected at least 14 days after participant's latest vaccine dose having values above the upper quantification limit of the test (33%, Fig. 2d). Anti-N antibody trajectories are shown in Supplementary Fig. S4.

To investigate how infection and vaccination history affect antibody levels, we fit kinetic models to individual-level antibody trajectories. Mean antibody rises were similar among age classes. Rises in anti-S binding antibody levels depended markedly on both infection and vaccination history (Fig. 3a, parameter estimates in Supplementary Table S2). The weakest estimated anti-S boost were the ones following infection in unvaccinated individuals, while the strongest boost was estimated following the first vaccine dose in previously infected

individuals. Among vaccinated and infected individuals, the estimated anti-S boost parameter decreased with the number of vaccine doses. Among uninfected individuals, the largest increase in anti-S levels occurred after the second vaccine dose, with similar levels for the first and third doses. Mean antibody half-lives showed less variation among boosting events, ranging from 50 days (95% CrI: 30–100) following the second vaccine dose in uninfected 18–64 y individuals to 510 days (140–1360) in 65+ individuals with two infections and one vaccine dose (Fig. 3b). Estimated antibody half-lives were similar across individuals both infected and vaccinated, regardless of the number of vaccine doses received. In turn, antibodies decayed faster in uninfected individuals following the second dose, as opposed to antibodies mounted with the first and third doses. Both boost levels and antibody decay rates had considerable, although uncertain, individual-level heterogeneity with coefficients of variation of 5.1 (95% CrI: 0.2–50.0) and 22.1 (0.3–65.5), respectively. These kinetic parameter estimates, along with inference on individual-level variability, allowed us to model antibody trajectories for each participant with a strong agreement with available serological measurements (Fig. 3c, Supplementary Fig. S5).

**Survival analysis during the Omicron exposure period**
In this second part of the analysis, we used survival analysis to evaluate the relationship between the projected anti-S binding antibody levels, as described above, and the hazard of infection during the Omicron exposure period (Fig. 1). Data on virologically confirmed infections during the exposure period (positive test or self-reported negative tests only, see "Methods") were available for 967 out of the 1083 participants, of whom we retained 900 with latest serology after April 1st, 2021 (Supplementary Fig. S1, Supplementary Table S3). The subsample included in this survival analysis was composed of 55% of female participants; 92% were younger than 65 years; 80% had received at least one vaccine dose prior to the start of the Omicron exposure period (December 25th, 2021); and 90% had at least one SARS-CoV-2 infection prior to the start of the exposure period (Supplementary Table S3). Out of these 900 participants, 227 had a virologically-confirmed infection during the Omicron exposure period. Self-reported symptoms information was available from questionnaire data for 219 infections out of the 227 reported during the exposure period. A total of 201/219 (91%) self-reported infections were accompanied by at least one symptom.

We found that the hazard of having an Omicron BA.1/BA.2 infection for individuals with anti-S binding antibody levels higher than a given arbitrary threshold, compared to those with levels below that threshold, decreased down to a minimum of a three-fold reduction in hazard at a threshold of 800 IU/mL (hazard ratio, HR 0.30, 95% CI: 0.22–0.41), and then plateaued for higher antibody level thresholds (Fig. 4a). In sensitivity analyses we found consistent effect sizes across antibody thresholds using logistic regression, as well as using different quantiles of the predicted antibody trajectories (Supplementary Material Section S6).

We, however, found that measured antibody levels do not have the same meaning in terms of a correlate of protection whether a participant had a history of infection or not, independently of antibody level (Fig. 4b). Similar proportions of Omicron infections were observed among vaccinated participants with no history of infection (and anti-N negative serology) irrespective of their anti-S antibody levels being below or above the 800 IU/mL threshold. Conversely, participants with a history of infection had a lower hazard of infection when having antibody levels above the 800 IU/mL threshold, regardless of their vaccination status (Fig. 4b, bottom row). Thus, for this anti-S antibody levels threshold, effect estimates stratified by infection and vaccination history showed no significant difference in hazard for uninfected (vaccinated) participants (HR 1.05, 95% CI 0.36–3.05), and a consistent hazard reduction for participants with a history of infection whether they were vaccinated (HR 0.30, 0.07–1.21) or not (HR 0.45,

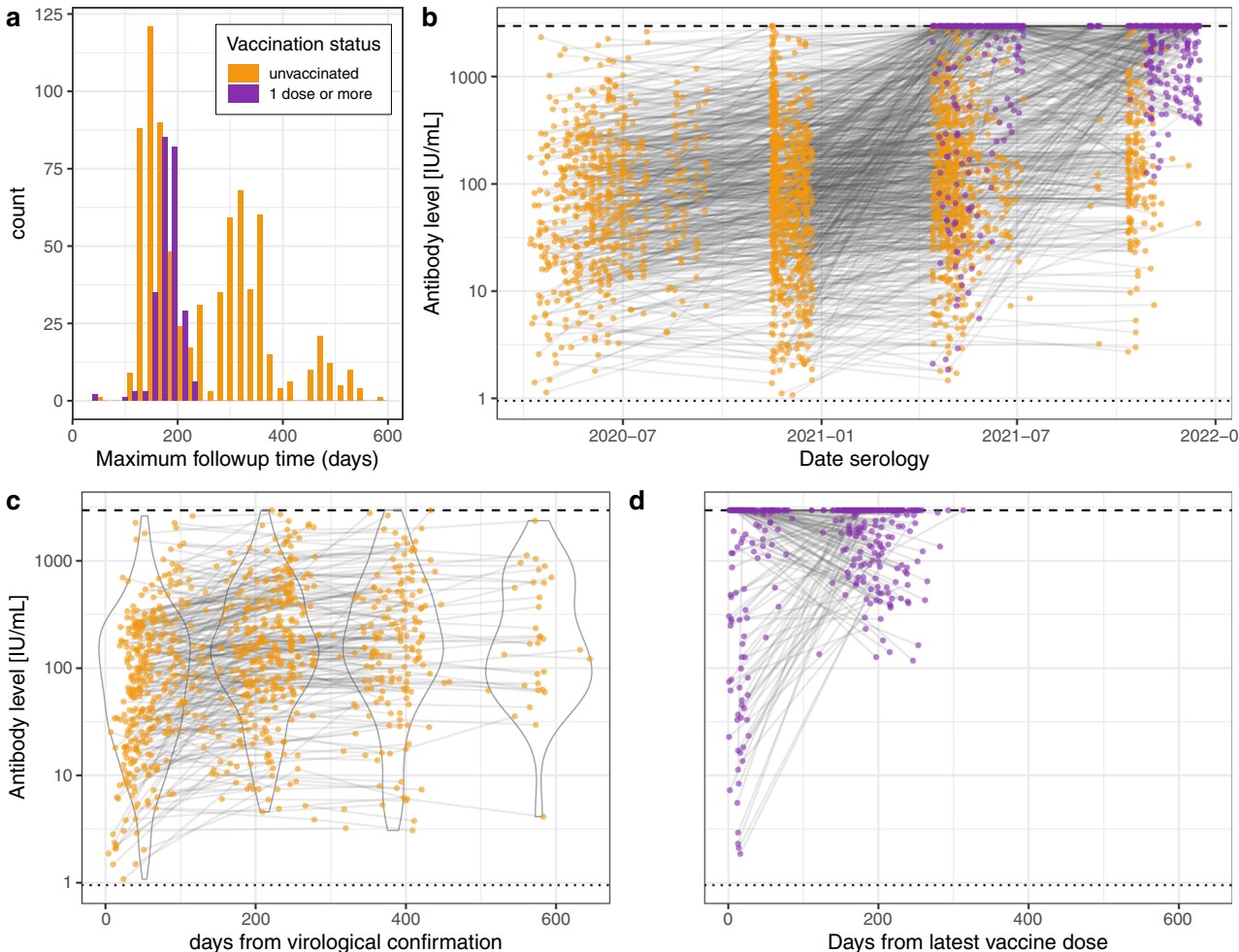

**Fig. 2 | Anti-S binding antibodies level trajectories. a** Follow-up time distribution (time from participant's first to last serology) for samples collected prior (n = 778, yellow) and post (n = 246, purple) vaccination when at least two positive samples were available. Note that participants may have multiple samples prior and post vaccination and may therefore appear in both categories. **b** Trajectories for all participants (n = 1083) by serological sampling date and according to vaccination status. Colors as in panel (**a**). **c** Trajectories of pre-vaccination samples by time from virological confirmation when available (n = 442), along with violin plots of antibody levels in discrete arbitrary categories of time post confirmation (0–149, 150–249, 250–449, 450+ days). **d** Trajectories post-vaccination by time from latest dose (n = 246). Dashed and dotted lines in panels **b**–**d** indicate the upper quantification limit (2500 U/mL, equivalent to 2960 IU/mL) and threshold for positivity (0.8 U/mL, equivalent to 0.95 IU/mL) of the test, respectively. Source data are provided as a Source Data file.

0.19–1.06), although small sample sizes in some of these categories yielded wide confidence intervals. Stratified results followed similar patterns for the other antibody thresholds with non-significant results for non-infected participants (Supplementary Fig. S8).

## Discussion

This longitudinal antibody study with follow-up times up to 20 months provided the opportunity to understand long-term anti-SARS-CoV-2 antibody dynamics and to evaluate binding antibody levels from a commercial widely available immunoassay as a correlate of protection against infections during the Omicron BA.1/BA.2 era. Anti-S antibodies persisted up to 20 months after the probable date of infection, with decay dynamics determined by infection and vaccination history. The strongest and longest-lasting antibody boosts occurred with vaccine doses following prior infection. Modeled antibody trajectories enabled the evaluation of binding antibody levels as a correlate of protection against Omicron BA.1/BA.2 infections, for which we found an overall three-fold reduction in the hazard of reporting a positive test for antibody levels above 800 IU/mL. Hazard reduction was, however, not observed for non-infected participants, indicating that the validity of anti-S binding antibody levels as correlates of protection for Omicron BA.1/BA.2 depends on infection history.

This study extends our previous work showing that anti-SARS-CoV-2 spike antibodies remain detectable after 22 months past probable infection as measured with the Roche anti-S immunoassay[22]. Our kinetic modeling results support previous findings indicating that antibody boost is strongest and longest lasting in vaccinees with a history of infection[19,23,24]. In contrast with previous findings, we found no significant difference in antibody boosting between age groups and slower decay rates in adults 65 years and older[20,25]. The slower decay rates may be due to age-specific differences in disease severity that we did not account for in these models, thus limiting the comparability of these findings with previous studies due to differences in disease severity profiles. Furthermore, we had a small number of participants over 65 years of age in our study, and these age-stratified results should be interpreted with caution. Finally, our results highlight the strong individual-level variability in antibody dynamics, which has been shown in previous antibody kinetic studies[25,26].

Survival analysis results on Omicron BA.1/BA.2 infections are in line with previous findings from vaccine trials targeting the ancestral strain and the Alpha variant, showing that binding antibody levels are an informative correlate of protection against SARS-CoV-2 infection[5,12]. These trials had found similar effect sizes of around a fivefold reduction in risk of Alpha infections at anti-S antibody levels of 600 IU/mL[12]

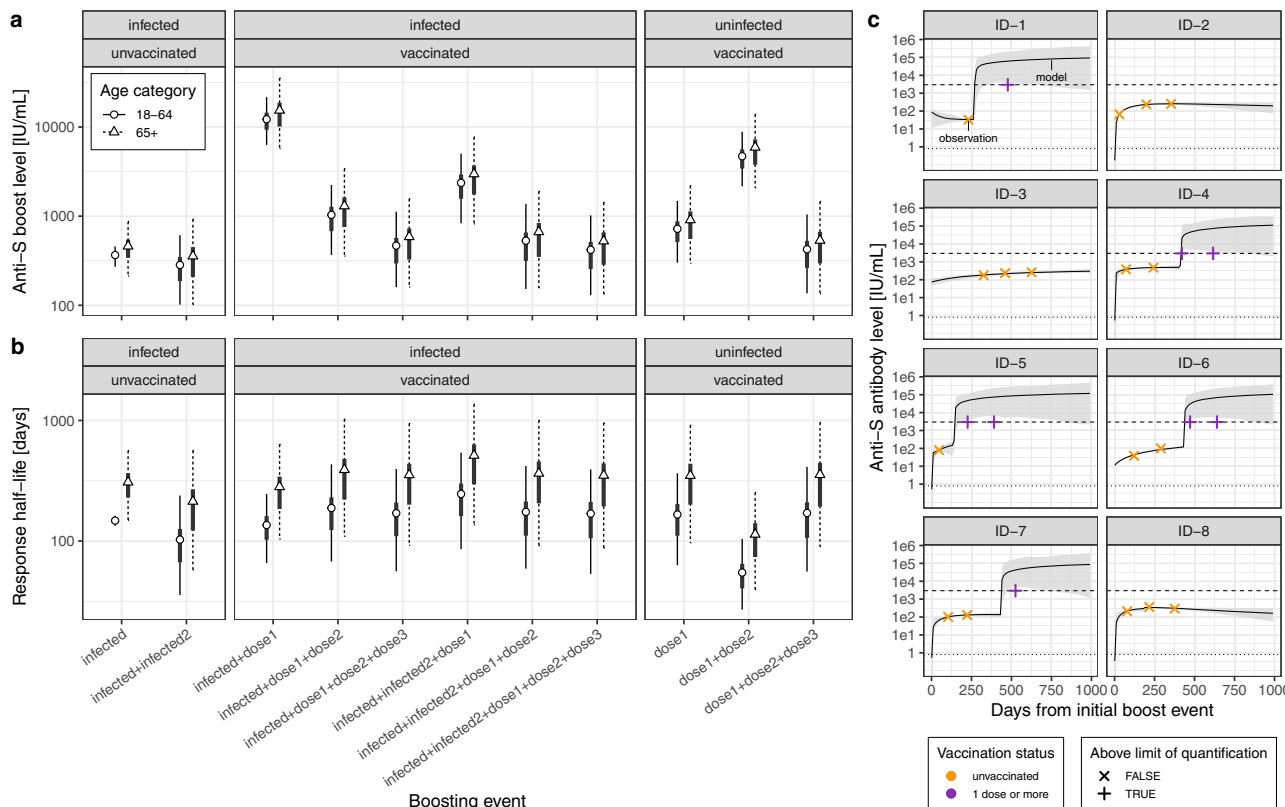

**Fig. 3 | Antibody dynamics inference. a** Inferred mean antibody level boosts following infection and/or vaccination by age category and infection/vaccination history (dots: mean, thick lines: 50% CrI, thin dotted/dashed lines: 95% CrI from 5000 posterior draws). "Dose1/2/3" denotes the vaccine dose, and "infected1/2" denotes the infection (first or second infection). Note that the order of boosting events is not taken into account in the model and that boosting events are considered to be additive. **b** Inferred mean antibody level half-lives with symbols as in panel (**a**). **c** Example of serological measurements and modeled antibody trajectories for a random set of participants (ID-1 to ID-8). Measurements were available before and/or after vaccination (colors) and were either below or above the Roche Elecsys anti-S upper quantification limit of 2960 IU/mL. Modeled trajectories are given in terms of the mean (line) and 95% CrI (shaded area, from 5'000 posterior draws). Source data are provided as a Source Data file.

and a halving of hazard by a 10-fold increase in anti-S titers for ancestral strain infections[5]. Moreover, our results are in line with available studies on Omicron BA.1/BA.2 subvariants, which have also found binding antibody levels to be correlates of protection against infection using in-house immunoassays[15–18]. In particular, one study using the same immunoassay found similar effect size estimates in a prospective study design using measured antibody levels, although not differentiating between infection/vaccination statuses and finding large confidence intervals[16]. On the other hand, we did not find differences in the hazard of having an Omicron BA.1/BA.2 infection with anti-S antibody levels below or above a certain threshold in the non-infected vaccinated group, as opposed to results reported for Delta infections[13]. Notably, this finding is supported by our recent work on neutralization capacity in the Geneva population[2]. Using the same immunoassay as in this study and a cell-free Spike trimer-ACE2 binding-based surrogate neutralization assay[8], we did not observe any significant correlation between anti-S binding and neutralizing antibody levels against Omicron subvariants in uninfected participants, as opposed to previously infected participants[2]. These results can be linked to growing evidence that hybrid immunity (infection plus vaccination) provides the strongest protection against Omicron subvariant infections[17,18,27]. This infection history-specificity thus warrants care in the interpretation of binding antibodies as correlates of protection against Omicron sublineages and could be immunoassay-dependent.

We note that it remains unclear whether these correlate of protection results extend to subsequent Omicron subvariants (BA.4, BA.5, BA.2.75, BQ.1, and others), which have been found, thanks to specific

mutations, to have stronger immune evasion capacity than the parent BA.1 strain[28,29]. Changes in immune evasion capacity may, theoretically if multiple mutations accumulate on the spike protein, impact the level of binding antibodies at which hazard reduction occurs, as well as its effect size. Moreover, our longitudinal serology follow-up was conducted before the circulation of the Omicron lineage in Geneva. The interpretation of anti-S antibody levels measured with this immunoassay following Omicron infections might need to be revisited in light of the evidence of reduced test sensitivity towards antibodies targeting the Omicron Spike protein[30].

This study has several limitations. Firstly, we only used the Roche Elecsys assay, which measures total anti-S antibodies (IgA/M/G), whose levels may correlate differently with overall immune function following infection or vaccination; other immunoassays may have different antibody binding characteristics. In particular, changes in the assay's antibody avidity with time since infection may impact the interpretation of antibody levels prior to infection related to functional immune capacity[10]. Secondly, analyses in the 65+ subgroup are limited by the small number of participants. Thirdly, our survival analysis to assess correlates of protection was based on modeled antibody trajectories and not on measurements at defined time points as done in studies available from vaccine trails[5]. Although modeled trajectories matched well with antibody participant-level measurements, the survival analysis results are subject to modeling uncertainty. While sensitivity analysis using the 2.5% and 97.5% prediction quantiles yielded qualitatively similar correlations of protection results, other sources of modeling uncertainty cannot be excluded. A further modeling limitation relates to the immunoassay's limit of quantification at the 1:10

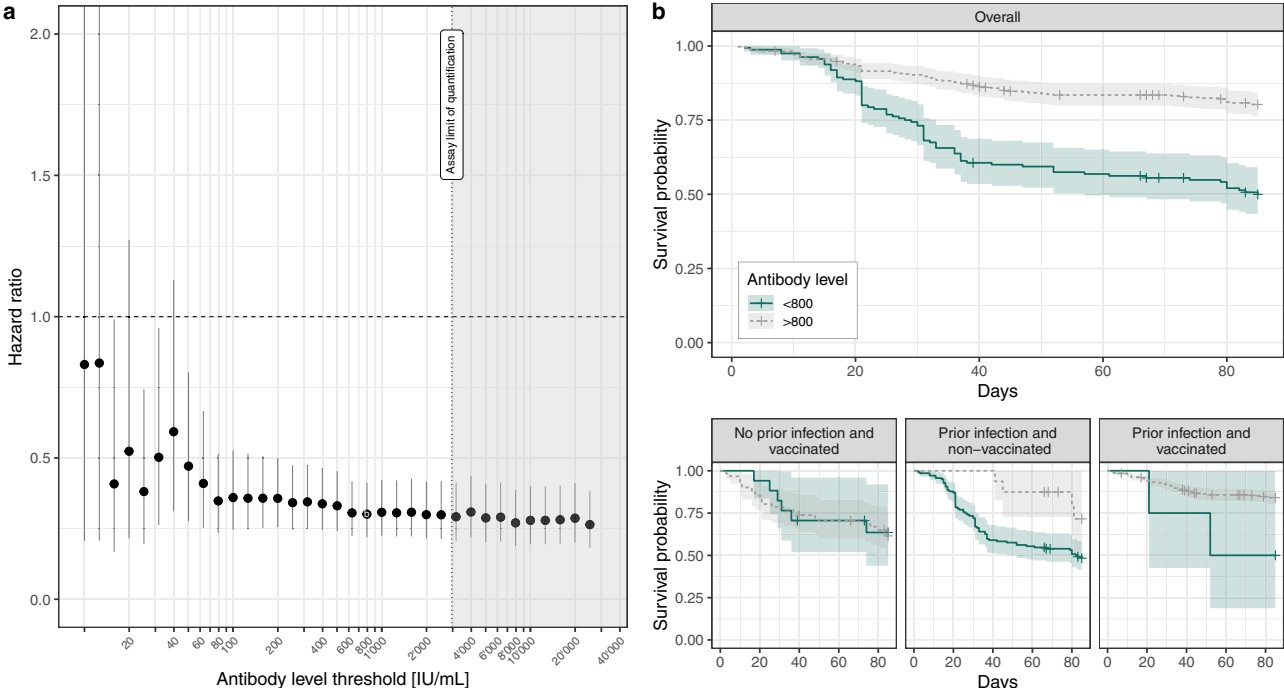

**Fig. 4 | Omicron BA.1/BA.2 infection survival analysis. a** Cox hazard ratio estimates based on proportional hazard models accounting for age and prior infection status across antibody level thresholds (dots give the mean, error bars give the 95% CI, n = 967 participants with available data, see Methods and Supplementary Fig. S1 for details on participants selection). **b** Kaplan–Meier curves of the probability of non-infection by SARS-CoV-2 Omicron BA.1/BA.2 stratified by whether predicted antibody levels during the exposure period were above or below 800 IU/mL, shown for the overall analysis dataset, and stratified by infection and vaccination history (ribbons indicate the 95% CIs). The selected value of 800 IU/ml corresponds to the threshold with the lowest hazard ratio in estimates in panel **a**. Day 0 corresponds to December 25th, 2021, when Omicron BA.1 accounted for more than 80% of infections in the state of Geneva (Fig. 1c). For this 800 IU/mL threshold, the overall sample size was of N = 562 (flowchart in Supplementary Fig. S1), subdivided into N = 78 for "No prior infection and vaccinated", N = 155 for "Prior infection and non-vaccinated", and N = 329 for "Prior infection and vaccinated". Source data are provided as a Source Data file.

dilution level used in this study, which for logistical constraints, could not be further diluted. We addressed this issue through censoring within our modeling framework. We further note that we could not account for unreported infections between the last serology for each participant and the exposure period, given that exclusion criteria for the survival analysis relied on reported infections. This may have resulted in the miss-classification of participants as falsely below the antibody threshold, thus leading to an underestimation of the reduction in infection hazard (i.e., bias towards the null). Fourthly, a large proportion of virologically confirmed infections (44%, 101/227) were self-reported as opposed to the other 56%, which were directly extracted from the state COVID-19 test registry (ARGOS). Reassuringly, of the 1083 participants in our longitudinal sample for whom tests in the registry were available, self-reported positive tests with matching dates were reported in 82% (491/599) of cases, thus suggesting a reasonable sensitivity of self-reporting. Moreover, we note that a proportion of Omicron infections during the exposure period were not reported due to subclinical infections, individual (self-)testing, and test reporting practices. As our outcome measure of reported infections is mainly composed of symptomatic infections, it may probably be more representative of more severe BA.1/BA.2 infections than infections across the clinical spectrum of the disease. Finally, both Omicron BA.1 and BA.2 subvariants circulated in the canton of Geneva during the study exposure period, and sequencing information on infection was not available, thus precluding a differential correlation of protection analysis for both subvariants.

Overall, this study extends findings against previous SARS-CoV-2 variants showing that anti-S binding antibody levels measured by a widely distributed immunoassay are a valid correlate of protection against Omicron BA.1/BA.2 infections. Importantly, we found that the validity of antibody levels as a correlate of protection depends on infection history as quantified with the immunoassay used in this study. Our results highlight the imperfect nature of protection after vaccination and/or infection. Even with perfect knowledge of infection and vaccination histories, inference about population-level immunity continues to pose challenges. Future studies may benefit from the modeling framework developed in this study to leverage longitudinal measurements to epidemiological outcomes. Taken together, these conclusions motivate further investigation of how immune landscape and immunoassay characteristics determine the interpretation of serological surveys into population levels of protection to inform public health decisions.

## Methods
### Study design
This study uses data from the population-based Specchio-COVID19 cohort, composed of adult participants recruited through serological surveys[31–34]. Following their baseline serology, participants in this cohort are regularly invited to complete online questionnaires, where they report SARS-CoV-2 test results, disease severity, and vaccination status and can be proposed one or several serological tests during the follow-up. Each participant coming for a follow-up serology provided a venous blood sample and filled in a short paper questionnaire on-site to update/complete their information on infection and vaccination statuses.

In this study, our main analysis consisted of two steps. Firstly, we analyzed antibody trajectories during the longitudinal serology period, when serological testing follow-up was conducted (April 6th, 2020 to December 17th, 2021). Secondly, we evaluated correlates of protection against infection during the "exposure period", using information on SARS-CoV-2 infections from the surge of the Omicron BA.1 subvariant in the state of Geneva until the end of the study period (December 25th, 2021 to March 20th, 2022) (Fig. 1). In this period there

were no specific quarantine and isolation measures following SARS-CoV-2 infection nor other specific recommendations in Geneva, which may have contributed to low virological testing rates. During the exposure period, Omicron BA.1 and BA.2 subvariants comprised nearly all infections. From all participants of the Specchio-COVID19 cohort, in this analysis, we only included those having at least two positive serologies and for whom we had complete vaccination information by March 20th, 2022 (Supplementary Material Fig. S1). During the study period, the only available COVID-19 vaccines in Switzerland were the mRNA-BNT162b2/Comirnaty from Pfizer/BioNTech (since December 2020), mRNA-1273 from Moderna/US NIAID (since January 2021), and the Janssen Ad26.COV2.S COVID-19 vaccine (since October 2021).

This study was approved by the Geneva Cantonal Commission for Research Ethics (CCER project number 2020-00881), and written informed consent was obtained from all participants.

## Immunoassays

For this study, we used the quantitative Elecsys anti-SARS-CoV-2 RBD immunoassay, which measures total antibodies (IgG/A/M) against the receptor binding domain of the virus spike (S) protein (#09 289 275 190, Roche-S, Roche Diagnostics, Rotkreuz, Switzerland). Seropositivity was defined using the cut-off provided by the manufacturer of ≥0.8 U/mL. Output test values were transformed to WHO international standard units by multiplying by a factor of 1.184. We calculated the intra-lot coefficient of variation (CV) for each batch of our internal positive control serum, and the maximum CV (7.3%) was used to define uncertainty in serological measurements in the kinetic model described below (Supplementary Material Section S2). To identify past infections in vaccinated participants, we also measured total levels of antibodies binding the nucleocapsid (N) protein using the semi-quantitative Elecsys anti-SARS-CoV-2 N immunoassay (#09 203 079 190, Roche-N). The three vaccines available in Switzerland during the study period elicit a response exclusively to the Spike protein of SARS-CoV-2, as opposed to infection, typically eliciting a response to both the N and S virus proteins. Although not the focus of the main analysis, we also present anti-N antibody trajectories in the Supplementary Material (Supplementary Material Section S3).

## SARS-CoV-2 virological tests data

For the correlation of protection analysis, results of PCR and antigenic tests were extracted from the ARGOS database up to March 20th, 2022. The ARGOS database consists of a general register of COVID-19 diagnostic tests performed in the state of Geneva since February 2020 and is maintained by the state directorate for health[35]. Data on test results from ARGOS were supplemented with additional information on COVID-19 diagnostic tests (PCR or antigen-based RDTs including self-tests) as self-reported by the participants through regular questionnaires.

## Statistical analyses

**Antibody trajectories analysis.** In the first step of the analysis ("longitudinal serology" period in Fig. 1), we characterized antibody dynamics by fitting the observed antibody trajectories to bi-phasic kinetic models[26]. These models assume an initial post-infection/vaccination antibody boost (increase in antibody levels at a given time post-exposure) followed by initially fast then slower exponential decay. We here expanded these models to account for multiple boosting events due to infection and/or vaccination. The size of antibody boosts and decay in time are determined by age, sex, and boosting history (the sequence of infections and vaccine doses). We further accounted for observed individual-level variability in anti-SARS-CoV-2 antibody kinetics. Inference was performed in a Bayesian hierarchical framework incorporating uncertainty of the timing of infection events in the absence of information on COVID-19 diagnostic tests. Model details are given in the Supplementary Material (Section S4).

**Survival analysis.** In the second step of the analysis, we evaluated binding antibody levels as a correlate of protection against Omicron BA.1/BA.2 infections during the "exposure period" (Fig. 1) using survival analysis methods. The aim was to infer the effect of being above different thresholds of binding antibody levels (as measured by the Roche-S immunoassay) on the hazard of confirmed SARS-CoV-2 infection during the exposure period. We focused on the exposure period from when Omicron BA.1 became dominant (more than 80% of samples on Dec 25th, 2021) up to the latest date for which we had access to the state registry of SARS-CoV-2 test results (March 20th, 2022). By this date, Omicron BA.2 had replaced Omicron BA.1 in Western Switzerland (90% vs. 10% of typed samples, Fig. 1). We excluded participants with either (a) no serology information after April 1st, 2021, or (b) uncertain infection status during the Omicron exposure period due to missing data (i.e., missing positive test result and missing self-reported absence of positive tests, details in Supplementary Material Section S1). We used Cox proportional hazards model, controlling for age and previous infection status based on our assumptions of the relationship between variables (Supplementary Material Section S5). Given that we used modeled antibody levels during the exposure period based on available serological prior measurements, we excluded participants for whom a boosting event (infection and/or vaccination) occurred between the last serological measurement and the start of the exposure period if the modeled antibody level was below the threshold of interest (Supplementary Material Section S1). We did not stratify estimates by a variant of infection prior to the exposure period due to small sample sizes. Potential informative censoring due to vaccination and/or infection during the Omicron exposure period was adjusted for through inverse probability weighting as implemented in the ipw R package[36].

## Data availability

This study included data from the state of Geneva's ARGOS database (Genecand, 2021)[35]. Data produced in this study can be made available to the scientific community upon submission of a data request application to the investigator's board via the corresponding author. All requests for data are responded within 3 months from submission. Source data are provided in this paper.

## Code availability

Stan model code and minimal testing datasets are available at https://github.com/UEP-HUG/serosuivi_2021_public.

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

## Acknowledgements

This study was funded by the Private Foundation of the Geneva University Hospitals and the General Directorate of Health of the canton of Geneva. We thank Aglaé Tardin and the team of the General Directorate of Health of Geneva for giving us access to the ARGOS state registry and the Hôpital de La Tour and the Clinique de Carouge for allowing us to use their premises for the recruitment of participants. We are deeply grateful to all the participants, without whom this study would not have been possible.

## Author contributions

J.P.S. and A.S.A. developed a methodology and led formal analysis as well as visualization. J.P.S. developed software, conducted the statistical analysis and created figures. M.E.Z., L.K., I.G., and S.S. conceptualized the project. M.E.Z. and S.S. validated results and led project administration. M.E.Z., J.L., and S.Y. contributed to the investigation. M.E.Z., J.L., R.D., and D.C. participated in data curation. O.K., N.V., and L.K. provided resources. I.G. and S.S. secured project funding. S.S. and A.S.A. supervised project execution. J.P.S., M.E.Z., I.G., S.S., and A.S.A. wrote the original draft. J.P.S., M.E.Z., J.L., S.Y., R.D., D.S.C., J.V., J.F.B., D.P., O.K., N.V., L.K., I.G., S.S., A.S.A. participated to the reviewing and editing of the article.

## Competing interests

The authors declare no competing interests.

## Additional information

[1]Unit of Population Epidemiology, Division of Primary Care Medicine, Geneva University Hospitals, Geneva, Switzerland. [2]Department of Epidemiology, Johns Hopkins Bloomberg School of Public Health, Baltimore, MD, USA. [3]Division of Laboratory Medicine, Department of Diagnostics, Geneva University Hospitals, Geneva, Switzerland. [4]General Directorate of Health, Geneva, Switzerland. [5]Division of Quality of Care, Geneva University Hospitals, Geneva, Switzerland. [6]Department of Medicine, Faculty of Medicine, University of Geneva, Geneva, Switzerland. [7]Infection Control Program and World Health Organization Collaborating Centre on Patient Safety, Geneva University Hospitals, Geneva, Switzerland. [8]Division of Internal Medicine, Hôpital de la Tour, Geneva, Switzerland. [9]Division of Infectious Diseases, Department of Medicine, Geneva University Hospitals, Geneva, Switzerland. [10]Geneva Centre for Emerging Viral Diseases, Geneva University Hospitals, Geneva, Switzerland. [11]Department of Health and Community Medicine, Faculty of Medicine, University of Geneva, Geneva, Switzerland. [12]Division and Department of Primary Care Medicine, Geneva University Hospitals, Geneva, Switzerland. [13]University Centre for General Medicine and Public Health, University of Lausanne, Lausanne, Switzerland. [23]These authors contributed equally: Idris Guessous, Silvia Stringhini, Andrew S. Azman. ✉e-mail: javier.perez@hcuge.ch

## the Specchio-COVID19 study group

Isabelle Arm-Vernez[3], Andrew S. Azman[1,2,23], Delphine Bachmann[14], Antoine Bal[1], Jean-François Balavoine[6], Rémy P. Barbe[15], Hélène Baysson[1], Julie Berthelot[1], Gaëlle Bryand-Rumley[1], François Chappuis[12], Prune Collombet[1], Sophie Coudurier-Boeuf[3], Delphine S. Courvoisier[4,5], Carlos de Mestral[1], Paola D'ippolito[1], Richard Dubos[1], Roxane Dumont[1], Nacira El Merjani[1], Antoine Flahault[16], Natalie Francioli[1], Clément Graindorge[1], Idris Guessous[11,12,23], Séverine Harnal[1], Samia Hurst[17], Laurent Kaiser[3,6,9,10], Omar Kherad[6,8], Julien Lamour[1], Pierre Lescuyer[3], Arnaud G. L'Huillier[18,19], Andrea Jutta Loizeau[1], Elsa Lorthe[1], Chantal Martinez[1], Ludovic Metral-Boffod[3], Mayssam Nehme[12], Natacha Noël[1], Francesco Pennacchio[1], Javier Perez-Saez[1,2]✉, Didier Pittet[6,7], Klara M. Posfay-Barbe[18,20], Géraldine Poulain[3], Caroline Pugin[1], Nick Pullen[1], Viviane Richard[1], Déborah Rochat[1], Khadija Samir[1], Hugo Santa Ramirez[1], Etienne Satin[12], Philippe Schaller[21], Stephanie Schrempft[1], Claire Semaani[1], Silvia Stringhini[1,13,23], Stéphanie Testini[1], Déborah Urrutia-Rivas[1], Charlotte Verolet[1], Pauline Vetter[9], Jennifer Villers[1], Guillemette Violot[22], Nicolas Vuilleumier[3,6], Ania Wisniak[1], Sabine Yerly[3] & María-Eugenia Zaballa[1]

[14]Hirslanden Clinique des Grangettes and Hislanden Clinique La Colline, Geneva, Switzerland. [15]Division of Child and Adolescent Psychiatry, Department of Woman, Child, and Adolescent Medicine, Geneva University Hospitals, Geneva, Switzerland. [16]Institute of Global Health, University of Geneva, Geneva, Switzerland. [17]Institute for Ethics, History, and the Humanities, Faculty of Medicine, University of Geneva, Geneva, Switzerland. [18]Division of General Pediatrics, Department of Woman, Child, and Adolescent Medicine, Geneva University Hospitals, Geneva, Switzerland. [19]Pediatric Infectious Diseases Specialist, Geneva University Hospitals and Faculty of Medicine, Geneva, Switzerland. [20]Department of Pediatrics, Gynecology & Obstetrics, Faculty of Medicine, University of Geneva, Geneva, Switzerland. [21]Arsanté, Organisation en soins, Geneva, Switzerland. [22]Communication Directorate, Geneva University Hospitals, Geneva, Switzerland.

