## [Peer Review File · Nature Communications]

Long term anti-SARS-CoV-2 antibody kinetics and correlate of protection against Omicron BA.1/BA.2 infectionREVIEWER COMMENTS

Reviewer #1 (Remarks to the Author):

In This manuscript Perz-Saez et al. utilize a large cohort of individuals to model antibody levels and predicted infection status. While the methods utilized appear sound in terms of antibody modeling and survival analysis, the manuscript lacks clear scientific framework for the conclusions drawn. The authors should clarify their grouping and sampling and tone down their overall conclusions based on the modeling and small sample size.

1. Many of the measures in this study are guesses or estimates. The antibody levels at the time of infection are modeled. The viral strain is estimated based on population level infection data.
2. Binding antibodies have not been shown to be a correlate of protection from vaccine or infection. There is no well-defined correlate, although neutralizing antibodies are suggestive.
3. There serological method used for anti-S here is not against the Omicron variant.
4. To have a clear correlate, one would need to know the antibody level at the time of infection. This is not the case here.
5. Out of 900 participants, only 227 had infections and there were diverse age, infection history and vaccine status. Modeling predicted antibody levels is not scientifically sound with this small of numbers.
5. If the modeled antibody levels were a true correlate, it would not matter if they were generated with a vaccine or infection. yet, the authors only observe significant differences with prior infection. This suggests that this is not a true correlate of protection, but is a correlate of the real correlate of protection. Which probably is neutralizing antibodies. Many studies have shown that binding antibodies can persist, while neutralizing antibodies decrease.
6. The authors do not show any correlation of antibody levels with decreased infection risk. Figure 4A looks like all detectable levels have some association with decreased infection. Why not 200IU or 1000 IU was selected?

Reviewer #2 (Remarks to the Author):

The authors report on data of 1,083 adult participants from a population-based cohort in Geneva, Switzerland, between Apr 2020 and Mar 2022, to evaluate the use of Spike binding antibodies as immune correlates of protection against infection using the Elecsys anti-SARS-CoV-2 immunoassay (Roche Diagnostics). Prior studies have indicated that binding antibodies are reasonable correlates of protection, and this study aims to assess if these prior findings continue to hold during the Omicron era (primarily BA.1 and BA.2). They do this by measuring S-antibody levels at multiple timepoints between Apr 2020 and Dec 2021, modeling antibody trajectories, and use Cox proportional hazard models to assessing differential risks of infection by modeled antibody levels during Omicron transmission. The key findings reported in this manuscript are that (i) S-antibody levels measured with this immunoassay correlate with protection against infection during predominantly BA.1/BA.2 transmission, (ii) antibody boost is strongest and longest lasting in vaccinees with a history of infection, (iii) there does not appear to be a significant difference in antibody boosting between age groups and slower decay rates in adults 65 years and older. This manuscript addresses an important question related to the use of binding antibodies for assessing immunological protection given three years into the pandemic methods for assessing population-immunity are poorly developed. Overall, the manuscript is mostly clear and well written. However, I have several comments and concerns as discussed below.

As the authors note, others have reported on binding antibody immune COP, but most were from the pre-Omicron era. A recent short communication assessed immune COP for BA.1, BA.2 and

BA.4/5, using a test negative design and the same Roche assay ([https://doi.org/10.1016/S1473-3099\(23\)00001-4](https://doi.org/10.1016/S1473-3099(23)00001-4)), but the current manuscript uses a fundamentally different study design, which is important to validate and triangulate COP estimates; and provides substantially more detail and additional analysis including differential boosting and response half-life stratified by antigen exposure type and sequence. Another preprint manuscript examines immune COP for BA.4/5 in the UK using the Oxford S-antibody assay (<https://doi.org/10.1101/2022.11.29.22282916>).

Comments regarding data & methodology:

1. Overall, the dataset used for these analyses is rich and of high quality and Statistics and treatment of uncertainty appears to be appropriately used.

2. The main approach of modeling antibody trajectories to estimate S-antibody levels at the time of infection is reasonable and is being used by other groups, although subject to some limitations. For example, it is unclear if undetected/unreported infection between the last serology sampling and the exposure period are considered. This is particularly relevant given not insignificant transmission was reported in the ~one month prior to the defined exposure period. This point should be clarified, median intervals between the last serology sampling and exposure period presented, and this limitation should be noted.

3. Similarly, despite that the authors clearly made efforts to capture acute infections, it is highly probable that a certain fraction of infections were not identified during the exposure period. This is more likely given (i) high rates of mild or asymptomatic infections with BA.1 and BA.2 infections, and (ii) overall higher population immunity that would further increase the likelihood of subclinical infections during the exposure period. This should be noted in the limitations. Relatedly, the number of infections detected during the exposure period should be reported.

4. In addition, and more importantly, how the authors modeled antibody trajectories when a relatively large proportion of the samples were above the upper limit of detection is unclear and should be clarified. Presumably, values $>2,500$ U/ml were analysed as 2,500 U/ml, although I do not see this stated. Given all key findings relate to antibody trajectory estimates, and given dilutions were only performed at the 1:10 ratio, the upper limit of quantification of 2500 U/mL will fail to fully capture antibody dynamics for a large fraction of study participants, as would be anticipated in a highly exposed population, and as demonstrated in Fig 2 (B, D). Additional dilutions for samples $>2,500$ U/ml would substantially strengthen the manuscript. Overall manufacturer (Roche) recommended dilutions are 1:10 to 1:100, therefore additional 1:10 dilutions would provide a measuring range to 25,000 U/ml, which should capture most if not all samples.

5. Fig 3 AB. The number of participants in each infection/vaccination category is unclear. For example, given there are only $N=95$ study participants ≥ 65 years, and given there are a total of 11 infection/vaccination categories, the number in each category would presumably be quite small and therefore generalizability limited. Please include details on the number of observations somewhere (supplementary materials such as S2 is fine). I do appreciate that limitations related to the 65+ year age category due to small sample size are noted in the discussion.

6. It is unclear why the threshold of 800 IU/ml was selected and if (and how much) survival findings would change if another threshold such as 400 or 1200 IU/ml were selected. Or, for example, if S-antibody levels were binned into multiple categories such as 0-399, 400-799, and 800+ IU/ml (or similar). In particular, the findings that the 800 IU/ml S-antibody threshold does not appear to be a predictor of risk of infection for those that were vaccinated but not previously infected (but is for the other categories), is surprising without any immediately obvious biological mechanism. As such, confirming that this finding holds when other thresholds are selected would strengthen this claim.

References are appropriate

Except for the comments above, the abstract, introduction and conclusions are appropriate.

Reviewer #3 (Remarks to the Author):

The work presented aims to identify correlates of protection against SARS-CoV2 Omicron BA1/BA2 infection based on analysis of pre-existing antibody levels (IgG/A/M) against the RBD of Spike viral protein. The analysis is based on repeated serological measurements on 1083 patients in Geneva between April 2020 and December 2021. An antibody kinetic model is used to estimate individual antibody levels in the subsequent period during the BA1/BA2 Omicron wave. Survival analysis is then conducted on a subgroup of participants to measure the risk of infection according to different thresholds of estimated antibody levels and history of exposure with infection, vaccination or both. Results reveal a 3-fold reduction in the hazard of having an infection with Omicron for participants having antibody levels above 800 IU/mL, regardless of their vaccination status. Regarding this infection history, it would be interesting to stratify participants according to the most probable cause of previous infection to account for a potentially different effect of the infecting variant (alpha or delta) on the following antibody responses and protection. The main limitations of this work are addressed in the discussion which makes this well-written paper transparent, rigorous, balanced and easy to understand for a large audience. One of the limitation of this work is the immunoassay that measures total anti-S antibodies (IgG/M/A). The measured values reflect a sort of cumulated signal for all antibodies that makes impossible to know their respective proportions and functional association with protection. Likely related to this, it is unusual to observe the total absence of seroreversion among participants during the long follow-up period, even for the categories of participants with lower estimated antibody half-lives. One might suspect that this multi-isotype assay cause an artificial inflation of antibody signal. As a consequence, the modelled antibody trajectories might be impacted more than just the standard modelling uncertainties. This particular issue together with the saturation of the signal in the upper quantification questions the validity of the measured and inferred antibody levels especially after several boosting events. It thus warrants care in the interpretation of antibodies as correlate of protection. It could be assay-dependent, as expressed but the authors. Whether specific to this study or transposable to another study context, the qualitative value and conclusions of this work remain relevant and of epidemiological significance since this commercial assay is widely distributed.

Response to Reviewers

Reviewer #1 (Remarks to the Author):

In This manuscript Perz-Saez et al. utilize a large cohort of individuals to model antibody levels and predicted infection status. While the methods utilized appear sound in terms of antibody modeling and survival analysis, the manuscript lacks clear scientific framework for the conclusions drawn. The authors should clarify their grouping and sampling and tone down their overall conclusions based on the modeling and small sample size.

We thank the reviewer for their critical evaluation of our work and their comments to frame the analysis and results. We believe that after addressing this and the other Reviewer's comments the manuscript is now better framed and its limitations are better explained.

1. Many of the measures in this study are guesses or estimates. The antibody levels at the time of infection are modeled. The viral strain is estimated based on population level infection data.

We agree that in the face of not having perfectly observed data we had to rely on estimates from our observed data. In response to the two specific examples provided, we don't believe either should be considered a 'guess,' and both are informed by measured data as illustrated below:

- Antibody levels used in the survival analysis are indeed modelled estimates based on established modeling frameworks for antibody kinetics which have been applied across pathogens including SARS-CoV-2 (Pelleau et al. 2021). Within our statistical modeling framework we propagate uncertainty in antibody measurements, infection times and kinetic parameters throughout the analyses to help ensure we appropriately capture uncertainty in the estimates. These estimates are subject to limitations inherent to this type of modeling approach, which we acknowledged in the Discussion section. Following comments by other Reviewers, we have now further stressed the limitation of modeling antibody kinetics above the assay's limit of quantification, which we have addressed following standard procedures in the field by using censoring in the model's likelihood function (Teunis et al., 2002).
- The assignment of SARS-CoV-2 variant was based on inferred proportions of variants in sequenced samples in the Geneva population. We acknowledge that having variant information on all PCR-tested participants would have been ideal. This is however unrealistic in population-based studies due to the large proportion of antigenic results that were included in the analysis and the fact that PCR samples were not systematically sequenced in Geneva, in particular during the Omicron waves. Variant assignment based on dominant variant circulation in time is therefore the best inference that can be attained, in particular due to the short windows of variant overlap. This approach has been adopted in other studies (Pulliam et al., 2022), for instance in that referred to by Reviewer 2 (Wei et al., 2022).

2. Binding antibodies have not been shown to be a correlate of protection from vaccine or infection. There is no well-defined correlate, although neutralizing antibodies are suggestive.

Evidence for binding antibodies as correlate of protection for vaccination and infection have been highlighted for Delta and Omicron BA.1 infections (Earl et al. 2021, Hertz et al. 2022, Zar et al. 2022), and have more recently been shown for BA.4/5 infections in references brought to our attention by Reviewer 2 (Wei et al. 2022, Nilles et al. 2023). We have added these references to the revised version of the manuscript.

3. There serological method used for anti-S here is not against the Omicron variant.

We acknowledged in the Discussion that the Roche-S immunassay has lower sensitivity for antibodies developed after Omicron infections than after wildtype infections (Springer et al. 2022). In this analysis, however, we use Roche-S levels to model antibody kinetics in the period prior to Omicron circulation in Geneva. Our results are therefore not affected by the lower sensitivity of the immunoassay to anti-Omicron SARS-CoV-2 antibodies. We however stressed in the limitations that extending our results to Roche-S binding antibody levels measured after the Omicron waves require caution.

4. To have a clear correlate, one would need to know the antibody level at the time of infection. This is not the case here.

We agree with this reviewer that in an ideal study design, antibodies would be measured a short time prior to infection. However, again this is unrealistic in population-based studies of this scale. In this study, we use all the information available over the study period to model antibody levels prior to infection. Of note, our estimates of antibody levels are based on a robust and established modeling framework for antibody kinetics. We accounted for uncertainty in modeled antibody levels in the sensitivity analysis in Figures S10, and our main results on the role of binding antibodies as correlates of protection remained valid when using the 2.5% and 95% posterior credible intervals of modeled antibody levels. Finally, our results are supported by a recent study brought to our attention by Reviewer 2 that found similar effect sizes, although with wider confidence intervals, for the same immunoassay but using measured antibody levels instead of modeled as in our case (Nilles et al. 2023).

5. Out of 900 participants, only 227 had infections and there were diverse age, infection history and vaccine status. Modeling predicted antibody levels is not scientifically sound with this small of numbers.

We acknowledged in the Discussion that our sample size is limited, and results warrant caution in their interpretation, in particular for the older age class. We however believe that our methodology using modeled antibody levels is scientifically well grounded as discussed above, and the posterior credible intervals account for sources of uncertainty in estimated antibody levels which we used in sensitivity analysis as mentioned in the point above. We also mention challenges in faced by our methodology in the limitations stressed in the Discussion. We further stress that we were interested to see that a recent study pointed by Reviewer 2 found similar effect sizes for the reduction in infection hazard with binding antibodies using the same immunoassay as in our study is supportive of the approach and results in our work (Nilles et al. 2023).

5. If the modeled antibody levels were a true correlate, it would not matter if they were generated with a vaccine or infection. yet, the authors only observe significant differences with

prior infection. This suggests that this is not a true correlate of protection, but is a correlate of the real correlate of protection. Which probably is neutralizing antibodies. Many studies have shown that binding antibodies can persist, while neutralizing antibodies decrease.

We agree that one of the important results of our analysis is that binding antibody levels as measured by this immunoassay for uninfected vaccinated participants was not associated to a reduced risk of Omicron BA.1/BA.2 infection, and is not correlated to neutralizing capacity as we have shown in a previous study (Zaballa et al. 2022). We however question the fact that binding antibodies are not a “true” correlate of protection, and it is unclear to us what it is meant by “true”. We could indeed argue that a correlate of a correlate of protection is itself a correlate of protection despite possibly being more distal in terms of immunological mechanisms, and that the notion of a true correlate of protection across the whole population is not generally the case (WHO, 2013). We believe that our analysis has value in showing that binding antibodies as measured by the widely used Roche-S immunoassay do serve a correlate of protection against Omicron BA.1/BA.2 in previously infected individuals. From a public health perspective, the main results of our analysis are particularly relevant given the high proportion of the general population having been previously infected by SARS-CoV-2 as highlighted by the latest serosurveys (Zaballa et al. 2022).

6. The authors do not show any correlation of antibody levels with decreased infection risk. Figure 4A looks like all detectable levels have some association with decreased infection. Why not 200IU or 1000 IU was selected?

The magnitude of the reduction in hazard ratio is indeed not linearly linked to binding antibody levels across the dynamic range of the immunoassay, but we do observe a decreasing trend with a plateau for higher BAU values. The choice of the threshold to illustrate our results was based on the lowest relative hazard rate. As detailed in the response to Reviewer 2, we have added the explanation of the choice of antibody threshold in the Results section and in the caption of Figure 4.

Reviewer #2 (Remarks to the Author):

The authors report on data of 1,083 adult participants from a population-based cohort in Geneva, Switzerland, between Apr 2020 and Mar 2022, to evaluate the use of Spike binding antibodies as immune correlates of protection against infection using the Elecsys anti-SARS-CoV-2 immunoassay (Roche Diagnostics). Prior studies have indicated that binding antibodies are reasonable correlates of protection, and this study aims to assess if these prior findings continue to hold during the Omicron era (primarily BA.1 and BA.2). They do this by measuring S-antibody levels at multiple timepoints between Apr 2020 and Dec 2021, modeling antibody trajectories, and use Cox proportional hazard models to assess differential risks of infection by modeled antibody levels during Omicron transmission. The key findings reported in this manuscript are that (i) S-antibody levels measured with this immunoassay correlate with protection against infection during predominantly BA.1/BA.2 transmission, (ii) antibody boost is strongest and longest lasting in vaccinees with a history of infection, (iii) there does not appear to be a significant difference in antibody boosting between age groups and slower decay rates in adults 65 years and older. This manuscript addresses an important question related to the use of binding antibodies for assessing immunological protection given three years into the pandemic methods for assessing population-immunity are poorly

developed. Overall, the manuscript is mostly clear and well written. However, I have several comments and concerns as discussed below.

As the authors note, others have reported on binding antibody immune COP, but most were from the pre-Omicron era. A recent short communication assessed immune COP for BA.1, BA.2 and BA.4/5, using a test negative design and the same Roche assay ([https://doi.org/10.1016/S1473-3099\(23\)00001-4](https://doi.org/10.1016/S1473-3099(23)00001-4)), but the current manuscript uses a fundamentally different study design, which is important to validate and triangulate COP estimates; and provides substantially more detail and additional analysis including differential boosting and response half-life stratified by antigen exposure type and sequence. Another preprint manuscript examines immune COP for BA.4/5 in the UK using the Oxford S-antibody assay (<https://doi.org/10.1101/2022.11.29.22282916>).

We thank this Reviewer for bringing to our attention these recent studies. We have added these to the Introduction and Discussion, mentioning in particular the study by Nilles et al., which used the same immunoassay:

L. 328-331: “In particular, one study using the same immunoassay found similar effect size estimates in a prospective study design using measured antibody levels prior to infection, although not differentiating between infection/vaccination statuses and finding large confidence intervals (Nilles et al., 2023).”

Comments regarding data & methodology:

1. Overall, the dataset used for these analyses is rich and of high quality and Statistics and treatment of uncertainty appears to be appropriately used.

We thank this Reviewer for the positive evaluation of our work.

2. The main approach of modeling antibody trajectories to estimate S-antibody levels at the time of infection is reasonable and is being used by other groups, although subject to some limitations. For example, it is unclear if undetected/unreported infection between the last serology sampling and the exposure period are considered. This is particularly relevant given not insignificant transmission was reported in the ~one month prior to the defined exposure period. This point should be clarified, median intervals between the last serology sampling and exposure period presented, and this limitation should be noted.

We thank the Reviewer for this point. To address this issue, we excluded participants for whom we had a reported infection between the last serology and the exposure period from the survival analysis, if the modeled antibody level was below the considered threshold (as described in Figure S1). We acknowledge however that unreported infections between the last serology and the exposure period could not be taken into account, and may affect our estimates. We however believe that miss-classification of participants above/below the antibody threshold would result in smaller effect sizes and larger confidence intervals. The fact that we see a signal despite this limitation can therefore be considered a conservative estimate.

We have added a figure to the supplement with the distribution of times from the exposure period to the latest serology (Figure S7 as shown below), and expanded the discussion to

mention the limitation of undetected infections between the latest serology and the exposure period in the revised manuscript:

L. 374-379: “We further note that we could not account for unreported infections between the last serology for each participant and the exposure period, given that exclusion criteria for the survival analysis relied on reported infections. This may have resulted in miss-classification of participants as falsely below the antibody threshold, thus leading to an under-estimation of the reduction in infection hazard (i.e., bias towards the null).”

3. Similarly, despite that the authors clearly made efforts to capture acute infections, it is highly probable that a certain fraction of infections were not identified during the exposure period. This is more likely given (i) high rates of mild or asymptomatic infections with BA.1 and BA.2 infections, and (ii) overall higher population immunity that would further increase the likelihood of subclinical infections during the exposure period. This should be noted in the limitations. Relatedly, the number of infections detected during the exposure period should be reported.

We agree that a proportion of infections may have gone unreported during the exposure period due to asymptomatic infections or voluntary testing. To respond to this point, we extracted self-reported symptoms for self-reported infections available in our questionnaires. A total of 219 answers were available out of the 227 infections reported during the exposure period (the remaining infections being reported through the State of Geneva’s centralized registry ARGOS, for which we didn’t have access to symptoms data). A total of 201/219 (91%) of participants reported having had symptoms. Our outcome of reported infections therefore likely corresponds to symptomatic infections. We have added this information in the results section, and discuss it in the limitations section of the Discussion:

L. 261-262: “Self-reported symptoms information was available from questionnaire data for 219 infections out of the 225 reported during the exposure period. A total of 201/219 (91%) self-reported infections were accompanied by at least one symptom.”

L. 385-390: “Moreover, we note that a proportion of Omicron infections were not reported due to subclinical infections, (self-)testing and test reporting practices. As our outcome measure of reported infections is mainly composed of symptomatic infection, it may be more representative of more severe symptomatic BA.1/BA.2 infections than infections across the clinical spectrum of disease.”

4. In addition, and more importantly, how the authors modeled antibody trajectories when a relatively large proportion of the samples were above the upper limit of detection is unclear and should be clarified. Presumably, values >2,500 U/ml were analysed as 2,500 U/ml, although I do not see this stated. Given all key findings relate to antibody trajectory estimates, and given dilutions were only performed at the 1:10 ratio, the upper limit of quantification of 2500 U/mL will fail to fully capture antibody dynamics for a large fraction of study participants, as would be anticipated in a highly exposed population, and as demonstrated in Fig 2 (B, D). Additional dilutions for samples >2,500 U/ml would substantially strengthen the manuscript. Overall manufacturer (Roche) recommended dilutions are 1:10 to 1:100, therefore additional 1:10 dilutions would provide a measuring range to 25,000 U/ml, which should capture most if not all samples.

We agree that additional dilution would provide a richer quantification of antibody trajectories, and strengthen the survival analysis. For logistical reasons, we are unfortunately incapable of performing an additional 1:10 dilution on the samples in this study. In the kinetic modeling framework, we account for the assay’s limit of quantification by modeling observations at the limit as censored as typically done in antibody kinetic studies (cf. Supplementary material), and the model draws information on all antibody trajectories to infer kinetics above the limit of quantification. This approach follows previous antibody kinetic modeling studies for SARS-CoV-2 (Pelleau et al. 2021). It is the resulting inferred antibody trajectories that were used as inputs in the survival analysis, using the posterior mean in the main analysis, and the 2.5% and 97.5% posterior quantiles in the sensitivity analysis (Figure S10).

We have now clearly stated this data and modeling limitation in the Discussion:

L372-374: “A further modeling limitation relates to the immunoassay’s limit of quantification at the 1:10 dilution level we used in this study which for logistical constraints could not be further diluted. We addressed through censoring within our modeling framework.”

5. Fig 3 AB. The number of participants in each infection/vaccination category is unclear. For example, given there are only N=95 study participants ≥65 years, and given there are a total of 11 infection/vaccination categories, the number in each category would presumably be quite small and therefore generalizability limited. Please include details on the number of observations somewhere (supplementary materials such as S2 is fine). I do appreciate that limitations related to the 65+ year age category due to small sample size are noted in the discussion.

We thank the Reviewer for this comment. We have now added the information on participants and infections in each infection/vaccination category in the Supplementary Material as Table S3 and cited this Table in the text when presenting Figure 3 as follows:

Table S3: Characteristics of participants retained for survival analysis during the period of Omicron exposure.

Characteristic	N = 900 ¹
Age	
[18,65)	827 (92%)
[65,Inf]	73 (8%)
Sex	
female	498 (55%)
male	402 (45%)
Prior infection	811 (90%)
Prior vaccination	716 (80%)
Prior infection and vaccination	
prior infection and vaccination	627 (70%)
prior infection only	184 (20%)
prior vaccination only	89 (10%)
¹ _n (%)	

6. It is unclear why the threshold of 800 IU/ml was selected and if (and how much) survival findings would change if another threshold such as 400 or 1200 IU/ml were selected. Or, for example, if S-antibody levels were binned into multiple categories such as 0-399, 400-799, and 800+ IU/ml (or similar). In particular, the findings that the 800 IU/ml S-antibody threshold does not appear to be a predictor of risk of infection for those that were vaccinated but not previously infected (but is for the other categories), is surprising without any immediately obvious biological mechanism. As such, confirming that this finding holds when other thresholds are selected would strengthen this claim.

Survival analysis results were produced for all thresholds and effect size estimates for each are presented in Figure 4A. We chose to show the Kaplan-Meyer curves for 800 BAU/mL because this was the smallest BAU value for which the hazard ratio was the lowest before the plateau at higher antibody levels.

We have made this explicit in the Results section:

L. 267 : “We found that the hazard of having an Omicron BA.1/BA.2 infection for individuals with anti-S binding antibody levels higher than a given arbitrary threshold, compared to those with levels below that threshold, decreased down to a **minimum** of a three-fold reduction in hazard at a threshold of 800 IU/mL (hazard ratio, HR 0.30, 95% CI 0.22-0.41)”,

and in the caption of Figure 4 :

“The selected value of 800 IU/ml corresponded to the threshold with the lowest hazard ratio in estimates in panel a.”.

We however confirm that results regarding the non-significance of reduced risk for vaccinated/uninfected individuals hold across thresholds. We have added this to the results section:

L. 286-288: “Stratified results followed similar patterns for the other antibody thresholds with non-significant results for non-infected participants (Supplementary figure S8).”,

as well as to the supplementary material with a new figure showing stratified estimates for all thresholds (Figure S8), as shown below.

Supplementary Figure S8: Stratified hazard ratio estimates across antibody thresholds. Estimates were stratified by previous vaccination and infection status as reported in the main text.

We agree that this result is particularly interesting, and confirms our previous finding that neutralizing antibodies against Omicron subvariants correlate poorly with binding antibody levels as measured by the Roche-S immunoassay in uninfected participants (Zaballa et al. 2022).

References are appropriate

Except for the comments above, the abstract, introduction and conclusions are appropriate.

We thank this Reviewer for the constructive comments that have improved the presentation of our work.

Reviewer #3 (Remarks to the Author):

The work presented aims to identify correlates of protection against SARS-CoV2 Omicron BA1/BA2 infection based on analysis of pre-existing antibody levels (IgG/A/M) against the RBD of Spike viral protein. The analysis is based on repeated serological measurements on 1083 patients in Geneva between April 2020 and December 2021. An antibody kinetic model is used to estimate individual antibody levels in the subsequent period during the BA1/BA2 Omicron wave. Survival analysis is then conducted on a subgroup of participants to measure the risk of infection according to different thresholds of estimated antibody levels and history of exposure with infection, vaccination or both. Results reveal a 3-fold reduction in the hazard of having an infection with Omicron for participants having antibody levels above 800 IU/mL, regardless of their vaccination status.

We thank this Reviewer for the time taken in the evaluation of our work.

Regarding this infection history, it would be interesting to stratify participants according to the most probable cause of previous infection to account for a potentially different effect of the infecting variant (alpha or delta) on the following antibody responses and protection.

We agree that variant of infection would provide interesting insight into the analysis. The sample size in our cohort is however too small to perform this analysis. We have added this point to Methods section:

l. 196-197: "We did not stratify estimates by variant of infection prior to the exposure period due to small sample sizes."

The main limitations of this work are addressed in the discussion which makes this well-written paper transparent, rigorous, balanced and easy to understand for a large audience.

We thank the Reviewer for the positive evaluation of our work.

One of the limitation of this work is the immunoassay that measures total anti-S antibodies (IgG/M/A). The measured values reflect a sort of cumulated signal for all antibodies that makes impossible to know their respective proportions and functional association with protection. Likely related to this, it is unusual to observe the total absence of seroreversion among participants during the long follow-up period, even for the categories of participants with lower estimated antibody half-lives. One might suspect that this multi-isotype assay cause an artificial inflation of antibody signal. As a consequence, the modelled antibody trajectories might be impacted more than just the standard modelling uncertainties. This particular issue together with the saturation of the signal in the upper quantification questions the validity of the measured and inferred antibody levels especially after several boosting events. It thus warrants care in the interpretation of antibodies as correlate of protection. It could be assay-dependent, as expressed but the authors.

We agree with this reviewer that the nature of the immunoassay used in this study poses challenges in the interpretation of the antibody level signal and its correspondence with immunological mechanisms influencing the probability of infection during the exposure period. We believe we had made these limitations explicit in the Discussion section of the manuscript. We have further stressed this point in the revised version:

L. 362-364: "In particular, changes in the assay's antibody avidity with time since infection may impact the interpretation of antibody levels related to functional immune capacity (L'Huillier et al., 2021)."

Whether specific to this study or transposable to another study context, the qualitative value and conclusions of this work remain relevant and of epidemiological significance since this commercial assay is widely distributed.

We thank the Reviewer for this comment, and have stressed the fact that the immunoassay is widely available and used:

L. 41-42: "to evaluate anti-Spike RBD antibody levels measured by a widely used immunoassay as a correlate of protection"

L. 98-99: "In this study, we aim to evaluate the use of a widely available and used immunoassay as a correlate of protection in the Omicron era."

L. 394-396: "Overall, this study extends findings against previous SARS-CoV-2 variants showing that anti-S binding antibody levels measured by a widely distributed immunoassay are a valid correlate of protection against Omicron BA.1/BA.2 infections."

References

Earle, K.A., Ambrosino, D.M., Fiore-Gartland, A., Goldblatt, D., Gilbert, P.B., Siber, G.R., Dull, P. and Plotkin, S.A., 2021. Evidence for antibody as a protective correlate for COVID-19 vaccines. *Vaccine*, 39(32), pp.4423-4428.

Hertz, T., Levy, S., Ostrovsky, D., Oppenheimer, H., Zismanov, S., Kuzmina, A., Friedman, L.M., Trifkovic, S., Brice, D., Chun-Yang, L. and Shemer-Avni, Y., 2022. Correlates of protection for booster doses of the BNT162b2 vaccine. *medRxiv*, pp.2022-07.

Nilles, E.J., Paulino, C.T., de St Aubin, M., Duke, W., Jarolim, P., Sanchez, I.M., Murray, K.O., Lau, C.L., Gutiérrez, E.Z., Ramm, R.S. and Vasquez, M., 2023. Tracking immune correlates of protection for emerging SARS-CoV-2 variants. *The Lancet Infectious Diseases*.

Pelleau, S., Woudenberg, T., Rosado, J., Donnadiou, F., Garcia, L., Obadia, T., Gardais, S., Elgharbawy, Y., Velay, A., Gonzalez, M. and Nizou, J.Y., 2021. Kinetics of the severe acute respiratory syndrome coronavirus 2 antibody response and serological estimation of time since infection. *The Journal of infectious diseases*, 224(9), pp.1489-1499.

Pulliam, J.R., van Schalkwyk, C., Govender, N., von Gottberg, A., Cohen, C., Groome, M.J., Dushoff, J., Mlisana, K. and Moultrie, H., 2022. Increased risk of SARS-CoV-2 reinfection associated with emergence of Omicron in South Africa. *Science*, 376(6593), p.eabn4947.

Springer, D.N., Perkmann, T., Jani, C.M., Mucher, P., Prüger, K., Marculescu, R., Reuberger, E., Camp, J.V., Graninger, M., Borsodi, C. and Deutsch, J., 2022. Reduced sensitivity of commercial Spike-specific antibody assays after primary infection with the SARS-CoV-2 Omicron variant. *Microbiology Spectrum*, 10(5), pp.e02129-22.

Teunis, P.F.M., Van Der Heijden, O.G., De Melker, H.E., Schellekens, J.F.P., Versteegh, F.G.A. and Kretzschmar, M.E.E., 2002. Kinetics of the IgG antibody response to pertussis toxin after infection with *B. pertussis*. *Epidemiology & Infection*, 129(3), pp.479-489.

Wei, J., Matthews, P.C., Stoesser, N., Newton, J., Diamond, I., Studley, R., Taylor, N., Bell, J., Farrar, J., Marsden, B. and Kolenchery, J., 2022. Correlates of protection against SARS-CoV-2 Omicron variant and anti-spike antibody responses after a third/booster vaccination or breakthrough infection in the UK general population. *medRxiv*, pp.2022-11.

World Health Organization, 2013. *Correlates of vaccine-induced protection: methods and implications* (No. WHO/IVB/13.01). World Health Organization.

Zar, H.J., MacGinty, R., Workman, L., Botha, M., Johnson, M., Hunt, A., Burd, T., Nicol, M.P., Flasche, S., Quilty, B.J. and Goldblatt, D., 2022. Natural and hybrid immunity following four COVID-19 waves: A prospective cohort study of mothers in South Africa. *EClinicalMedicine*, 53, p.101655.

Zaballa, M.E., Perez-Saez, J., de Mestral, C., Pullen, N., Lamour, J., Turelli, P., Raclot, C., Baysson, H., Pennacchio, F., Villers, J. and Duc, J., 2023. Seroprevalence of anti-SARS-CoV-2 antibodies and cross-variant neutralization capacity after the Omicron BA. 2 wave in Geneva, Switzerland: A population-based study. *The Lancet Regional Health—Europe*, 24.

REVIEWERS' COMMENTS

Reviewer #1 (Remarks to the Author):

The authors have addressed the comments of the reviewers and have strengthened the manuscript. However, many of the limitations the authors have now addressed in the discussion do not increase the impact of the findings. The authors made little attempt to improve the experimental or statistical findings. Simply stating that the authors can not do the work does not improve the findings. It is unclear the utility of this work for broader or current SARS-CoV-2 studies because of the use of modeling both antibody kinetics and viral information. However, the limitations are addressed and the methodology may be useful for future viral-host response research.

Reviewer #2 (Remarks to the Author):

The authors have addressed my concerns and, although limitations remain, these have been reasonably addressed as such in the revised manuscript.

Reviewer #3 (Remarks to the Author):

Thanks to the authors who have taken time to reply in details to the various comments I and other reviewers made.

These responses are very satisfying to me and reinforce the positive evaluation I already had during first round of review.

The limitations of this work are well addressed in the Discussion and the additional revisions or detailed explanations through main text edits or Supplementary material further increase quality of the manuscript.